# A DSSC Electrolyte Preparation Method Considering Light Path and Light Absorption

**DOI:** 10.3390/mi13111930

**Published:** 2022-11-09

**Authors:** Jianjun Yang, Jiaxuan Liu, Yaxin Li, Xiaobao Yu, Zichuan Yi, Zhi Zhang, Feng Chi, Liming Liu

**Affiliations:** 1College of Electron and Information, University of Electronic Science and Technology of China Zhongshan Institute, Zhongshan 528402, China; 2School of Optoelectronic Science and Engineering, University of Electronic Science and Technology of China, Chengdu 610054, China

**Keywords:** electrolyte, DSSC, UV-VIS absorption spectra, transmittance

## Abstract

The electrolyte is one of the key components of dye-sensitized solar cells’ (DSSC) structure. In this paper, the electrolyte formulation of a new DSSC with external photoanode structure was studied. Based on the idea that the electrolyte should match the light absorption and light path, iodine series electrolytes with different concentrations were configured and used in the experiment. The results showed that the external photoanode structure solar cells assembled with titanium electrode had the best photoelectric conversion ability when the concentration of I_2_ was 0.048 M. It achieved the open circuit voltage of 0.71 V, the short circuit current of 8.87 mA, and the filling factor of 57%.

## 1. Introduction

Dye-sensitized solar cells are mainly composed of photoanodes, counter electrodes, and electrolytes. The photoanode is mainly composed of light-absorbing dyes and conductive electrodes [1,2,3]. After the attached dye of the photoanode absorbs the emitted light, the separation of electron hole pairs occurs. The electrolyte is catalyzed by the counter electrode, resulting in an oxidation–reduction reaction, which makes the components of the DSSC return to the original state. The whole process generates current flowing through the external circuit to realize the conversion of light energy to electric energy [4,5,6,7]. The three parts have different functions, so researchers have been used to conducting independent research on these three parts of DSSC [8,9,10,11,12,13].

The location of the electrolyte as the redox reaction taking place has a great influence on the performance of DSSC. DSSC electrolytes include quasi-solid, solid electrolytes [14,15,16,17,18,19,20], and liquid electrolytes [21,22,23,24,25,26,27,28,29,30], of which liquid electrolytes are the most commonly used. Although liquid electrolytes have had problems such as solvent volatilization and electrode leakage corrosion [20,21], the preparation of solid and quasi-solid electrolytes have been complex and costly.

In the traditional back illuminated dye-sensitized solar cells, the incident light only needs to pass through a thin layer to reach the photoanode. However, for some dye-sensitized solar cells, the influence of the light absorption characteristics of the colored electrolyte on the photoelectric conversion performance of the cell needs to be considered. For example, the dye-sensitized solar cell with the external photoanode structure [31] adopted in this paper has the structure shown in the Figure 1. Therefore, it is of great significance to study a kind of liquid electrolyte to optimize light absorption.

## 2. Experiments

### 2.1. Experimental Materials

Adding 4-tert-butyl pyridine (TBP) to the liquid electrolyte could improve the open circuit voltage (Voc) [26,27,28], while guanidinium thiocyanate (GuSCN) was used as a Voc improver and a short-circuit photocurrent (Jsc) enhancer [28,29,30]. Therefore, the iodine electrolyte used in this paper consists of solvent: acetonitrile, solute: guanidine thiocyanate, iodine monomer, lithium iodide, PMII and TBP. The relevant information of each component is shown below.

LiI (purity of >99%), iodine, I_2_ (purity of >99%), PMII (purity of >99%), TBP (purity of >99%), GuSCN (purity of >99%), and the dye N719 (purity of >99%) were purchased from Sigma Aldrich. The absolute ethanol (purity of >99%) and acetone (purity of >99%) were purchased from Zhongshanjingke Company, Zhongshan, Guangdong, China.

### 2.2. Electrolyte Preparation

Different concentrations of electrolyte were used in the experiments. The highest concentration of electrolyte was prepared first, and then different amounts of solvents were added dropwise to obtain different concentrations of electrolyte. Taking the electrolyte with I_2_ concentration of 0.06 M as an example, the preparation method of electrolyte is described below.

The highest concentration of electrolyte preparation:

Firstly, we weighed 1.523 g I_2_, 8.031 g LiI, and 0.708 g GuSCN, then placed them into the measuring cylinder with a small amount of acetonitrile, placed the measuring cylinder into the balance, added 6.761 g TBP and 7.563 g PMII in turn with a rubber tipped dropper, removed out the measuring cylinder, added acetonitrile to the measuring cylinder, stirred while making the addition, and when it was almost 100 mL, used a dropper to add 100 mL. We transferred the solution in the measuring cylinder to a clean beaker with a magnetic rotor, sealed the beaker mouth with cling film, and put the beaker on a magnetic stirrer for 20 min.

2.Preparation of different concentrations:

In order to obtain the electrolyte with different concentrations, we divided 100 mL of electrolyte prepared in step (1) above into ten 10 mL small samples, numbered 1–10. We diluted the electrolyte in label 1 to 30 mL to obtain electrolyte with I_2_ concentration of 0.02 M. For the rest, we added acetonitrile dropwise to dilute the electrolyte in each vial to a concentration to be explored during the experiment, such as I_2_ of 0.03 M, 0.04 M, and 0.05 M.

### 2.3. Characterization

The ultraviolet visible spectra were obtained by an ultraviolet visible spectrometer (UV-24500, Avantes, Appeldom, The Netherlands). The J-V curve of DSSC was obtained by the combination of OAI solar simulator (LCSS150, Zolix, Beijing, China) and KEITHLEY2400 digital source meter (Oriel 94023A, Newport). The external quantum efficiency of DSSC was obtained by the quantum efficiency test system (PEC-S20, Gifu, Japan).

## 3. Results and Discussion

### 3.1. Light Absorption Characteristics

#### 3.1.1. Study on the Absorption Characteristics of Solute

The influence of different solutes on the light absorption characteristics of the electrolyte was studied by using an ultraviolet visible spectrometer. By studying the absorption spectrum of each solute, the components that had a great impact on the transparency of the electrolyte were analyzed.

The absorption spectrum of the dye N719 solution used in the external photoanode structure is shown in Figure 2a. It can be seen that the dye molecule had absorption peaks at 310 nm, 380 nm, and 530 nm in the wavelength band above 300 nm.

Preparation was carried out for 0.05 M I_2_ acetonitrile solution, 0.5 M LiI acetonitrile solution, 0.5 M TBP acetonitrile solution, 0.06 M GuSCN acetonitrile solution and 0.3 M PMII acetonitrile solution separately. After diluting different solutes 30 times with acetonitrile in proportion, we measured the ultraviolet visible absorption spectrum of each solute with an ultraviolet visible spectrometer, and the results are shown in Figure 2b.

Through the absorption spectra, it can be intuitively observed that in the wavelength band above 300 nm, only iodine had an obvious absorption peak for visible light, while LiI, TBP, GuSCN, and PMII had almost no absorption in the wavelength band above 300 nm. Comparison of the dye absorption spectrum shows that the electrolyte also had a strong absorption near 380 nm, that is, I_2_ was the component in the electrolyte that had the greatest impact on the liquid transmittance.

#### 3.1.2. Study on Different Concentrations Absorption Characteristics

By controlling the concentrations of iodine in the electrolyte, the effect of concentrations on the absorption characteristics of the electrolyte was explored. According to Lambert Beer law:(1)A=lg(1/T)=Kbc

*A* was the absorbance, *T* was the transmittance that was the incident light intensity on the transmitted light intensity ratio, and *K* was the molar absorption coefficient, which was related to the nature of the absorbing material and the wavelength of the incident light λ, where ***c*** was the concentration of the light absorbing substance, and *b* was the thickness of the absorption layer. It could be inferred that the higher the concentration of iodine, the stronger the absorption of sunlight. This inference was verified by experiments.

According to the electrolyte formula described in part 2, the electrolyte was prepared with iodine concentrations of 0.02 M, 0.03 M, 0.04 M, 0.05 M, and 0.06 M, respectively, diluted at 30 times at the same time, and then the UV-VIS spectrometer was used to test again. The UV-VIS absorption spectra of several electrolytes are shown in Figure 3.

It can be seen from Figure 3 that with the increase in electrolyte solute concentrations, its absorption of sunlight also increased significantly, which was consistent with the inference.

### 3.2. Photoelectric Performance

In the traditional back illuminated dye-sensitized solar cells, the incident light only needs to pass through a thin layer to reach the photoanode, and the light absorption has little effect on the performance of the cell. Because the electrolyte concentration affects the light absorption, the photoelectric performance of the external photoanode structure in this paper will be affected accordingly.

In order to comprehensively consider the influence of electrolyte on the photoelectric performance of external photoanode structure cells, electrolyte was injected with different concentrations into semi-finished solar cells with the same specifications and materials. Then, the J-V characteristics of the solar cells with different concentrations of electrolyte were tested. The J-V characteristic curves of the solar cells were combined with five concentrations of electrolyte in the same figure, as shown in Figure 4.

As it can be seen intuitively from Figure 4, when the iodine concentration was 0.05 M, the DSSC with external photoanode structure had the best J-V characteristics, and when the concentration was low or high, its photoelectric characteristics were significantly reduced.

It can be seen from Figure 5 that the short-circuit current density of the solar cells changed most obviously with the change in concentrations. When the concentration range was 0.02 M to 0.06 M, the minimum short-circuit current density was only 3.3 mA, and the maximum short-circuit current density was 8.62 mA. The open circuit voltages did not change much. The voltages of the DSSC with different concentrations of electrolyte were always between 0.6 V and 0.7 V, and the filling factor also changed significantly. It can be seen intuitively from the J-V curve that at low concentrations, The curve of the DSSC tended to be a straight line, which also meant lower filling factor and photoelectric conversion efficiency.

### 3.3. Electrolyte Influence Mechanism

In this paper, six groups of electrolytes with different concentrations were set up using the same formula of electrolyte, and FTO electrode DSSC with the same traditional sandwich structure were injected respectively, and then packaged. By measuring the photoelectric performance of the traditional structure and comparing it with the external photoanode structure, the mechanism of the influence of the electrolyte on the external photoanode structure was explored.

Comparing the external quantum efficiency of the device to the FTO device, as shown in Figure 6, the external quantum efficiency was significantly lower at the two peaks of 310 nm and 380 nm at short wavelengths, which also corresponded to the conclusion of the absorption spectrum of the electrolyte. The iodine-based electrolyte prepared in this paper had obvious absorption peaks at short wavelength bands around 290 nm and 370 nm, it could be judged that the concentrations of electrolyte had a significant effect on the photoelectric conversion efficiency of external photoanode dye-sensitized solar cells.

In this paper, the graph of the short-circuit current density of the external titanium photoanode structure solar cells and the traditional FTO solar cells with the change in concentrations was shown in Figure 7.

It can be seen from Figure 7 that when the electrolyte used in this paper was used in the traditional FTO structure, it was almost unchanged with the concentrations’ change, and the short-circuit current density value rose steadily with the concentrations’ increase, and tended to be stable after 0.06 M. However, when it was used for the external photoanode structure, according to the above, with the increase in electrolyte concentrations, its light absorption would also increase, and the light transmittance of the electrolyte and its hole transmission capacity would restrict each other with the change in concentrations, affecting the short circuit current of the external photoanode structure. According to the conversion formula of absorbance and transmittance, the logarithm curve of transmittance versus concentrations at 380 nm of electrolyte was also drawn in the figure.

As shown in Figure 7, the short-circuit current density of the external photoanode structure reached the maximum value with the concentration near 0.05 M, and the current density decreased with the continuous increase in the electrolyte concentrations. Before the peak point, the enhancement of hole transmission capacity caused by the increase in concentrations was dominant. The light absorption caused by the increase of concentrations after the peak point was dominant. The loss of incident light caused by electrolyte was even greater, and the stable short-circuit current density continued to decline.

### 3.4. Study on More Efficient Electrolyte

From the above performance changes in the DSSC under different electrolyte concentrations, among the three indicators of open circuit voltages, short-circuit current density, and filling factor, the most obvious change with concentrations was the short-circuit current density. The five points where the short-circuit current density changed with electrolyte concentrations were fitted in Origin software using third-order polynomial. The results are shown in Figure 8.

The function formula of the fitted curve was (2), where *E* was 10, the square of the determination coefficient r of the fitted curve was as high as 0.99, and the sum of the squares of the residuals was 0.063.
(2)$$Jsc=E^{4}\left(-39.33c^{3}+3.89c^{2}-0.1c\right)+10.94$$

Found the first derivative of the function to obtain:(3)$$Jsc^{\prime }=E^{4}\left(-117.99c^{2}+7.78c-0.1\right)$$

Found the second derivative:(4)$$Jsc^{\prime \prime }=E^{4}\left(-235.98c+7.78\right)$$

According to the first derivative and the second derivative of the short-circuit current density function obtained above, it was calculated that the concentrations of I_2_ had a maximum point (0.048, 8.8) in the interval of [0.02, 0.06].

As the concentration gradient set in the above experiment was large, four groups of electrolyte concentration experiments with small gradients were added in the range of 0.04 M to 0.06 M to explore a more accurate electrolyte concentration range, which was compared with the fitting results of Origin software.

By dropping different amounts of solvent acetonitrile into the 0.06 M iodine elemental electrolyte, electrolytes with iodine elemental concentrations of 0.044 M, 0.048 M, 0.052 M, and 0.056 M were obtained, and their J-V characteristics were tested again. The results are shown in Figure 9.

The short-circuit current density, open circuit voltage, and filling factor of electrolytes with different iodine concentration were measured. The results are shown in Figure 10.

It can be seen from Figure 10 that as the concentrations of iodine increased according to the gradient of 0.004 M, the open circuit voltages and short-circuit current density peak in the range of 0.04 M to 0.05 M. Among them, the short-circuit current density changed greatly. At the concentration of iodine at 0.048 M, it reached a maximum of 8.87 mA, which was basically consistent with the extreme points (0.048, 8.8) fitted by Origin software in the previous text. Therefore, it could be further determined that the fitting curve Formula (2) could more accurately express the relationship between the electrolyte concentrations and the short-circuit current density. The open circuit voltages always changed little, hovering between 0.6 V and 0.7 V, and reached the maximum value of 0.71 V at 0.048 M. The change in filling factor with concentrations was also similar to that in the large gradient experiment described above, and it was always maintained at around 50% to 60%. Therefore, it can be seen that the optimal concentrations of electrolyte for dye-sensitized solar cells with external photoanode structure on titanium substrate was I_2_ at 0.048 M. However, with the continuous increase in solute concentrations, the short-circuit current density had a downward trend. According to the change in electrolyte transmittance with the increase in concentrations described above, it could be inferred that this was one of the main reasons for the low short-circuit current density.

## 4. Conclusions

In this paper, firstly, through the UV-VIS spectra, it was confirmed that the iodine element in the electrolyte had the greatest impact on the solar absorption, and its concentrations had a great impact on the photoelectric effect of the solar cells, and the appropriate concentration range was determined. Secondly, by comparing the traditional structure with the external photoanode structure, the mechanism of the influence of electrolyte on the external photoanode structure was obtained. Before the peak point, the enhancement of hole transmission capacity caused by the increase in concentrations was dominant; the light absorption caused by the increase in concentrations after the peak point was dominant. Finally, the most suitable electrolyte was determined by using the Origin software fitting and verified by the experiment, which was consistent with the Origin fitting result, that is, the external photoanode structure DSSC assembled with titanium electrode had the best photoelectric conversion capability when the concentration of I_2_ was 0.048 M, and could achieve the open circuit voltage of 0.71 V, the short circuit current of 8.87 mA, and the filling factor of 57%.

## Figures and Tables

**Figure 1 micromachines-13-01930-f001:**
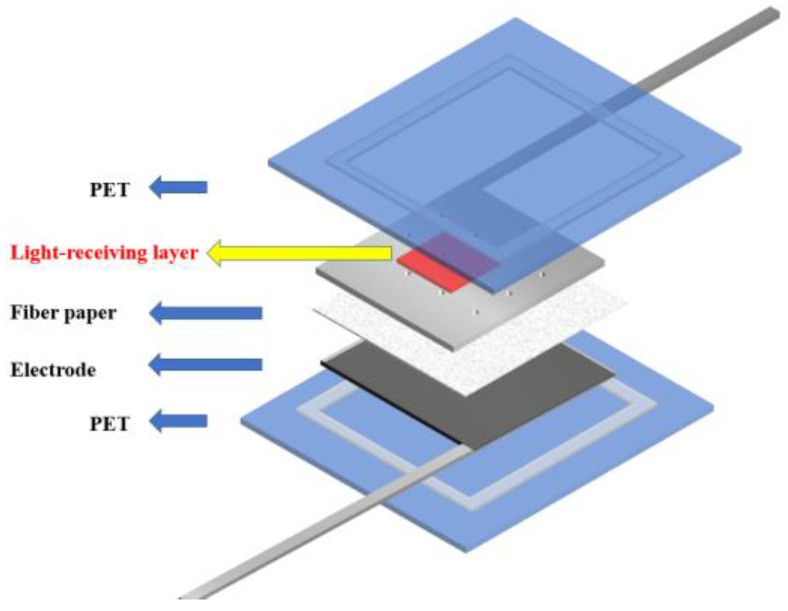
DSSC with external photoanode structure.

**Figure 2 micromachines-13-01930-f002:**
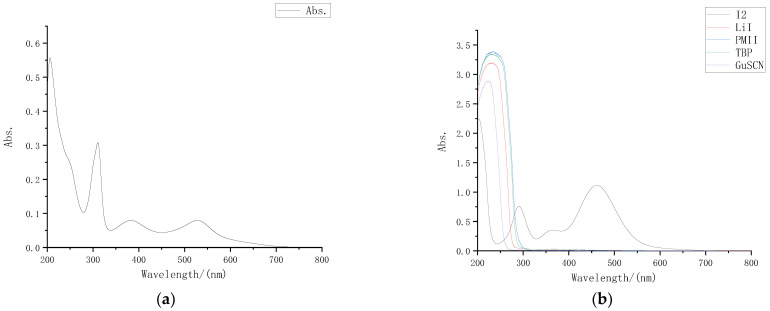
UV-VIS absorption spectrum: (**a**) UV-VIS absorption spectrum of N719 dye; (**b**) UV-VIS absorption spectra of different solutes.

**Figure 3 micromachines-13-01930-f003:**
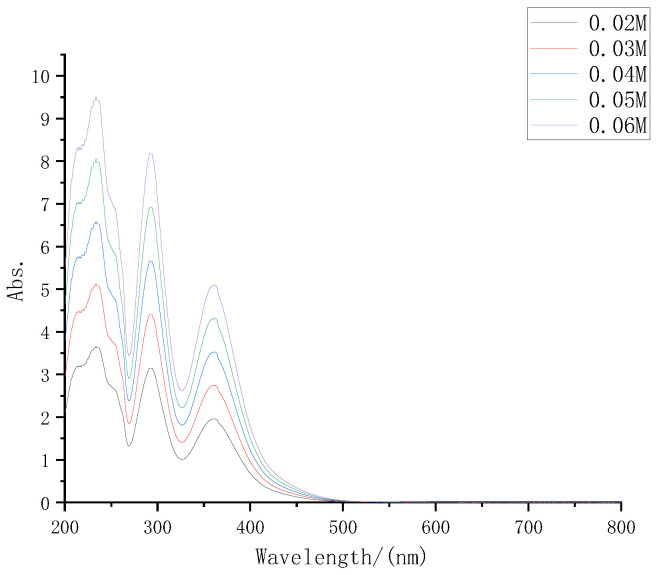
UV-VIS absorption spectra of iodine electrolyte with different concentrations.

**Figure 4 micromachines-13-01930-f004:**
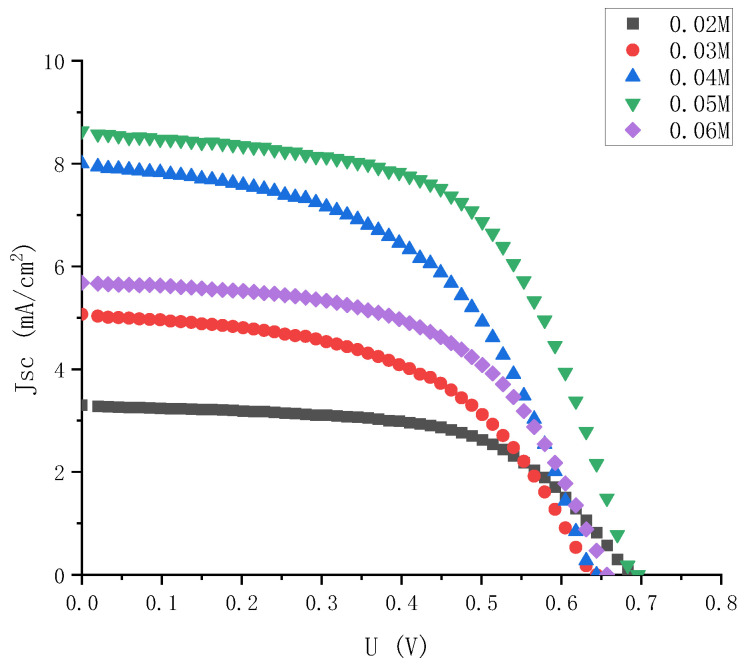
J-V characteristic curve of electrolyte solar cells with high gradient and different iodine concentrations.

**Figure 5 micromachines-13-01930-f005:**
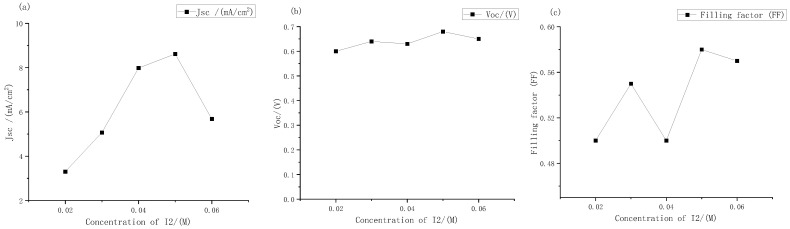
Performance changes in the external photoanode structure DSSC with different concentrations of iodine under large gradient: (**a**) short circuit current density; (**b**) open circuit voltage; (**c**) filling factor.

**Figure 6 micromachines-13-01930-f006:**
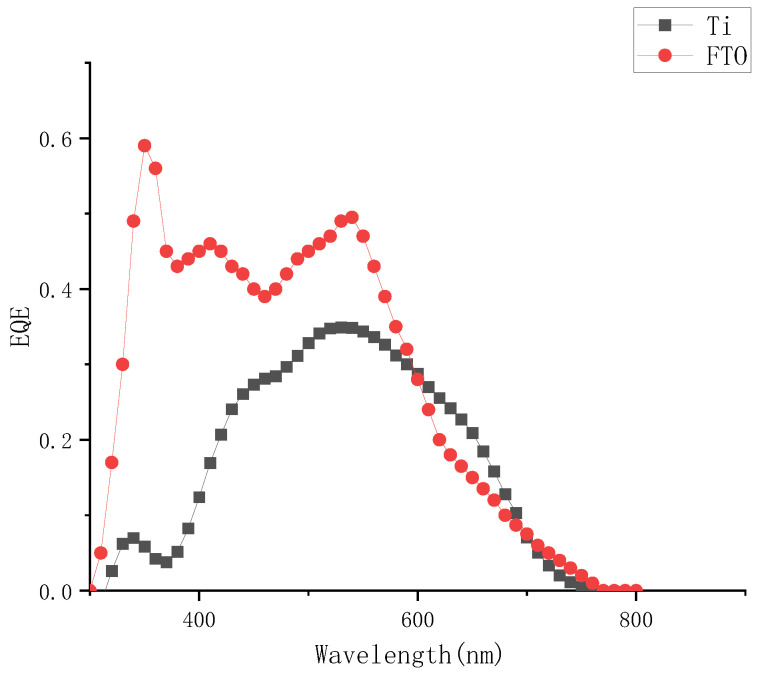
External quantum efficiency curve of different solar cells when the concentration of I_2_ was 0.05 M.

**Figure 7 micromachines-13-01930-f007:**
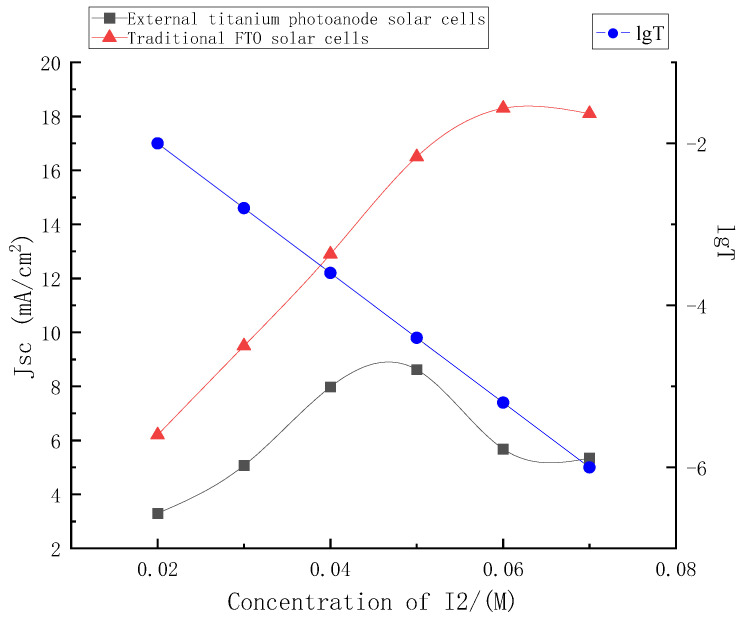
Change curve of short-circuit current density between traditional FTO solar cells and external titanium photoanode solar cells.

**Figure 8 micromachines-13-01930-f008:**
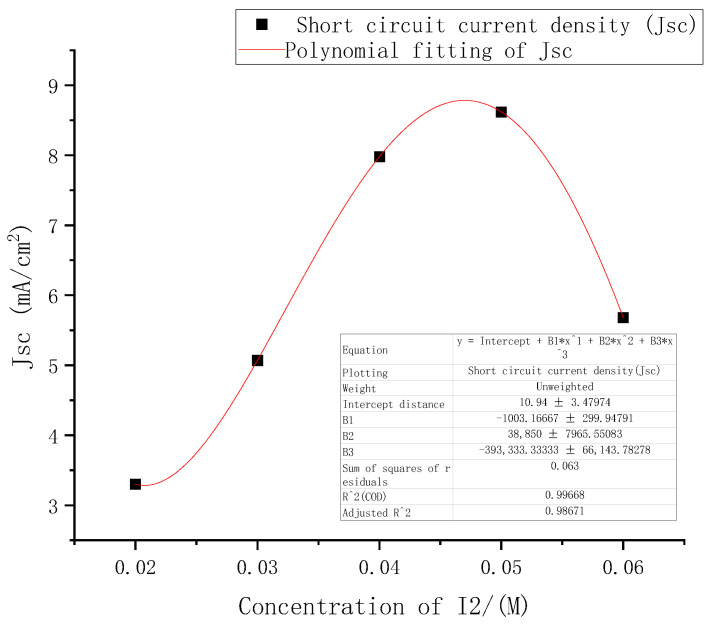
Fitting curve of short circuit current density at different electrolyte concentrations.

**Figure 9 micromachines-13-01930-f009:**
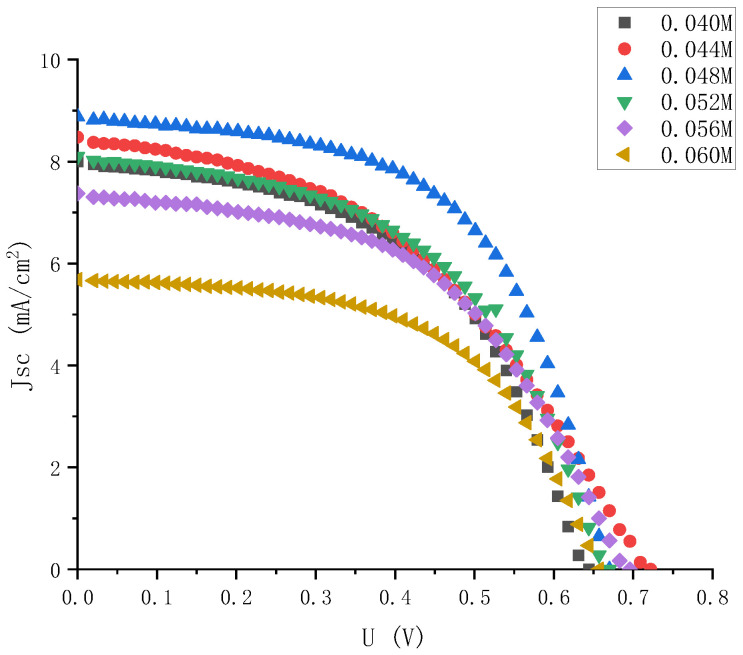
J-V characteristic curve of electrolyte solar cells with small gradient and different iodine concentrations.

**Figure 10 micromachines-13-01930-f010:**
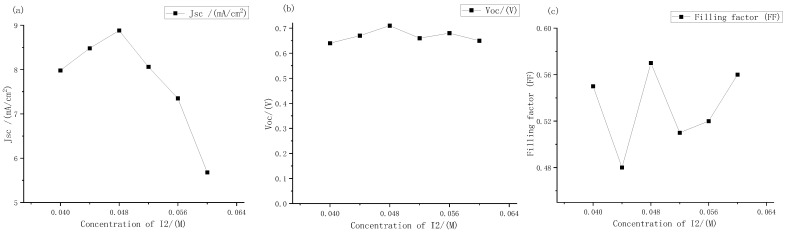
Performance changes of the external photoanode structure DSSC with different concentration of iodine under large gradient: (**a**) short circuit current density; (**b**) open circuit voltages; (**c**) filling factor.

## Data Availability

Data are contained within the article.

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
