# Peer review of "A DSSC Electrolyte Preparation Method Considering Light Path and Light Absorption"

_micromachines, 2022, doi:10.3390/mi13111930_

Round 1

Reviewer 1 Report

The current manuscript reports the component modification of the electrolyte of dye-sensitized solar cells. They got the best concentration of I2 in the electrolyte. However, about DSSC, there were a lot of papers discussing the related issues. The FF of the current DSSC is much lower. So, I don't recommend its publication in its current form. 

Author Response

Thanks for your questions. Please refer to the submission.

Reviewer 2 Report

1.      What is the reason for choosing different concentrations of I2 in this research work? Already a lot of researchers informed about the optimized I2 concentration.

2.      Noticed absorption plot of I2 concentration is very high in Fig 3.  Then how it satisfies Beer-Lambert law?

3.      Why in solar cell author mentioned Battery? It is not fit here.

4.      How the author fabricated the solar cell is not discussed in the Ms?

5.      What is the size of the cell is also not discussed.

6.      How many cells are fabricated for the single concentration of electrolyte?

7.      Stability of the cells is important compared to the optimized I2 concentration mentioned by researchers earlier in the literature.

8.      Electrochemical characterizations of the cell are missing.

9.      Calculate Isc from the EQE plot and compare the results.

10.  The scope of the work is to be clearly highlighted in the MS.

11.  Improve the abstract properly.

12.  What is the exact role of tBP in boosting the performance of DSSC?

13.  There are a few grammatical errors in the manuscript. Rectify it.

14.  The unit for the concentration of electrolyte are mentioned wrongly in some places

Author Response

(The authors gave the same response as above.)

Round 2

Reviewer 1 Report

The manuscript is well-revised.

Reviewer 2 Report

Accepted as such